# Epidemiology, etiology, and diagnosis of health care acquired pneumonia including ventilator-associated pneumonia in Nepal

**Sabina Dongol[1], Gyan Kayastha[2], Nhukesh Maharjan[1], Sarita Pyatha[1], Rajkumar K. C.[1], Louise Thwaites[3], Buddha Basnyat[1], Stephen Baker[4], Abhilasha Karkey[1]\***

**1** Patan Academy of Health Sciences, Patan Hospital, Oxford University Clinical Research Unit, Kathmandu, Nepal, **2** Patan Academy of Health Sciences, Patan Hospital, Kathmandu, Nepal, **3** The Hospital for Tropical Diseases, Wellcome Trust Major Overseas Programme, Oxford University Clinical Research Unit, Ho Chi Minh City, Vietnam, **4** Cambridge Institute of Therapeutic Immunology & Infectious Disease (CITIID), Department of Medicine, University of Cambridge, Cambridge, United Kingdom

\* akarkey@oucru.org

**Data Availability Statement:** All relevant data are within the manuscript and its Supporting Information files.

## Abstract

Epidemiologic data regarding health care acquired pneumonia (HAP) and ventilator-associated pneumonia (VAP) from Nepal are negligible. We conducted a prospective observational cohort study in the intensive care unit (ICU) of a major tertiary hospital in Nepal between April 2016 and March 2018, to calculate the incidence of VAP, and to describe clinical variables, microbiological etiology, and outcomes. Four hundred and thirty-eight patients were enrolled in the study. Demographic data, medical history, antimicrobial administration record, chest X-ray, biochemical, microbiological and haematological results, acute physiology and chronic health evaluation II score and the sequential organ failure assessment scores were recorded. Categorical variables were expressed as count and percentage and analyzed using the Fisher's exact test. Continuous variables were expressed as median and interquartile range and analyzed using Kruskal-Wallis rank sum test and the pairwise Wilcoxon rank—sum test. 46.8% (205/438) of the patients required intubation. Pneumonia was common in both intubated (94.14%; 193/205) and non-intubated (52.36%; 122/233) patients. Pneumonia developed among intubated patients in the ICU had longer days of stay in the ICU (median of 10, IQR 5–15, P< 0.001) when compared to non-intubated patients with pneumonia (median of 4, IQR 3–6, P< 0.001). The incidence rate of VAP was 20% (41/205) and incidence density was 16.45 cases per 1,000 ventilator days. Mortality was significantly higher in patients with pneumonia requiring intubation (44.6%, 86/193) than patients with pneumonia not requiring intubation (10.7%, 13/122, p<0.001, Fisher's exact test). Gram negative bacteria such as *Klebsiella* and *Acinetobacter* species were the dominant organisms from both VAP and non-VAP categories. Multi-drug resistance was highly prevalent in bacterial isolates associated with VAP (90%; 99/110) and non-VAP categories (81.5%; 106/130). HAP including VAP remains to be the most prevalent hospital-acquired infections (HAIs) at Patan hospital. A local study of etiological agents and outcomes of HAP and VAP are required for setting more appropriate guidelines for management of such diseases.

**Funding:** This project and AK was funded as a leadership fellow through the Oak Foundation (OCAY-15–547). URL:https://oakfnd.org/ The funders had no role in study design, data collection and analysis, decision to publish, or preparation of the manuscript.

**Competing interests:** The authors have declared that no competing interests exist.

## Introduction

Pneumonia is clinically defined as the presence of a new lung infiltrate with evidence that the infiltrate is triggered by an infectious agent such as, the new onset of fever, purulent sputum, or leukocytosis [1]. Healthcare acquired pneumonia (HAP) is an infection of the pulmonary parenchyma that develops>48 hours of admission to a health care facility and is commonly caused by pathogens that circulate in hospital settings [2]. In clinical practice, HAP is suspected when a patient presents with fever, impaired oxygenation, and suppurative secretions [3]. HAP is an important infectious disease worldwide and is associated with high morbidity, mortality, and additional health system expenditure [4]. In the US, the prevalence of HAP has been estimated to be1.6% of all hospital admissions, representing a rate of 3.63 cases per 1,000 patient-days [5]. Epidemiologic data regarding HAP in Asia are scarce; however, the incidence of HAP is predicted to be high across Asia and especially problematic in intensive care units (ICUs), where the proportion of ICU-acquired respiratory infections ranges from 9% to 23% of admissions [6].

Ventilator-associated pneumonia (VAP) is a subtype of HAP that develops in ICU patients who have been mechanically ventilated for at least 48 hour [2, 7, 8]. VAP remains one of the most common infections in patients requiring invasive mechanical ventilation and is the leading cause of ICU mortality [2, 7]. The reported prevalence of VAP vary from 5 to 40% of ventilated patients depending on country, ICU type, and criteria used to diagnose VAP [9]. In high-income countries, a combination of surveillance, education, and tailored intervention and prevention bundles have led to a major reduction in VAP [10]. However, even with the implementation of such programs, VAP is still commonly reported in the US [7]. In Asia there are limited data on incidence of VAP, the causative pathogens, and their antimicrobial susceptibility profiles [6, 11].A meta-analysis which encompassed 88 studies from 22 Asian countries from 2008 to 2018 indicated that the pooled incidence density of VAP in low-middle-income countries (LMICs) (18.5 per 1,000 ventilator-days) was more than twice that in high-income countries (9.0 per 1,000 ventilator-days) [12].

VAP has received little attention in LMICs until relatively recently [13]. In a low income country, like Nepal, where the incidence of infectious disease is high and strategies for control and prevention are weak, the opportunity for nosocomial infection is significantly higher [11, 14–16].This problem is further exacerbated by antimicrobial resistance (AMR) in organisms such as *Acinetobacter baumannii* and *Klebsiella pneumoniae*, which are responsible for a large proportion of nosocomial infections and commonly multi-drug resistant (MDR) [17, 18].

A delayed diagnosis and delay in initiating appropriate therapy in VAP may be associated with poor outcomes [2, 19–21]. Therefore, an early and accurate diagnosis is fundamental in the management of patients with VAP. In order to develop effective therapeutic strategies to optimize the use of antimicrobial agents we need a better understanding of the local pathogens causing. Therefore, we performed a prospective study to describe some epidemiological features of HAP among patients admitted to the ICU of major tertiary hospital in Kathmandu, Nepal. We measured the incidence rate of VAP, investigated the antimicrobial susceptibility profiles of the etiological agents, and compared clinical profiles associated with HAP/VAP mortality.

## Materials and methods

### Ethics approval and consent to participate

This study was approved by Nepal Health Research Council (NHRC) (Reference number 11/2016,Date: 11 March 2016) and Oxford Tropical Research Ethics Committee (OxTREC 32–16,

Date:19 October 2016). Adult patients admitted in the ICU or next-of-kin of the patient were approached for written informed consent to participate in this study.

## Setting and study design

This was a prospective observational cohort study conducted in the ICU of Patan hospital between April 2016 and March 2018. Patan hospital is a 450-bed tertiary care referral teaching hospital with 15 ICU beds, located in the Lalitpur Metropolitan area of the Kathmandu valley in Nepal.

## Study structure

All adult patients, $\geq$ 18 years of age admitted to the ICU were eligible to participate in the study. Adult patients admitted in the ICU or next-of-kin of the patient were approached for written informed consent to participate. Patients who denied consent and under the 18 years of age were not included in the study. Upon recruitment, demographic data, medical history, anti-microbial administration record, chest X-ray or other imaging findings, biochemical, microbiological and haematological results, and clinical parameters were recorded in a case report form (CRF). The acute physiology and chronic health evaluation (APACHE) II score and the sequential organ failure assessment (SOFA) score were recorded from the biochemical findings of the day of admission. Daily observation of the individual was conducted and CRF completed until an outcome of discharge, death, transfer to another ward or development of VAP.

VAP was defined by following the modified US Centers for Disease Control and Prevention criteria which requires to fulfill radiographic, systemic, and pulmonary criteria [16, 22–24] "Fig 1".

The day when the patient fulfilled the criteria of VAP was taken as day 0 of VAP diagnosis. These VAP confirmed patients were followed up on day 3, day 7, and day14. During these visits comparable clinical information was collected via hospital records. The final follow up was conducted on day 30 by phone if the patient was discharged, or in person if the patient was still in the hospital. Final diagnosis or working diagnosis if still under admission were recorded in the patient CRF.

## Sample collection for microbiological culture

Respiratory samples [either tracheal aspirates (TA), bronchoalveolar lavage (BAL), or sputum] and blood samples were obtained from enrolled patients for microbiological culture. A respiratory sample of either TA or BAL was obtained from all patients before the diagnosis of VAP. The decision for BAL or TA samples were at the discretion of the treating physician.

## Collection of TA sample

TA samples were collected as previously described, following local standard operating procedures [25]. Specimens were transported to the microbiology laboratory, and processed within 2 hours of collection. The tracheal aspirate specimens were examined by Gram staining, and the aspirate fluid was diluted 1:1 with Sputasol (Oxoid) and incubated at 37˚C, with periodic agitation, until liquefaction. The sample was diluted (1:1, $10^{-1}$ and $10^{-2}$) using maximum recovery diluent (Oxoid), and 20 ml 1:1 diluent was inoculated onto blood agar and chocolate agar plates. Additionally, 20μl of the $10^{-1}$ and $10^{-2}$ dilutions was plated onto MacConkey media and blood agar base (Mast diagnostics, UK). Inoculated media were incubated at 37˚Cand examined after 24 and 48 h of incubation. The threshold used to discriminate between infection and colonization was $\geq 1 \times 10^{5}$ colony forming unit (CFU)/ ml⁻1 (i.e., 20 colonies on either media from the $10^{-2}$ dilution). Colonies above this threshold were identified

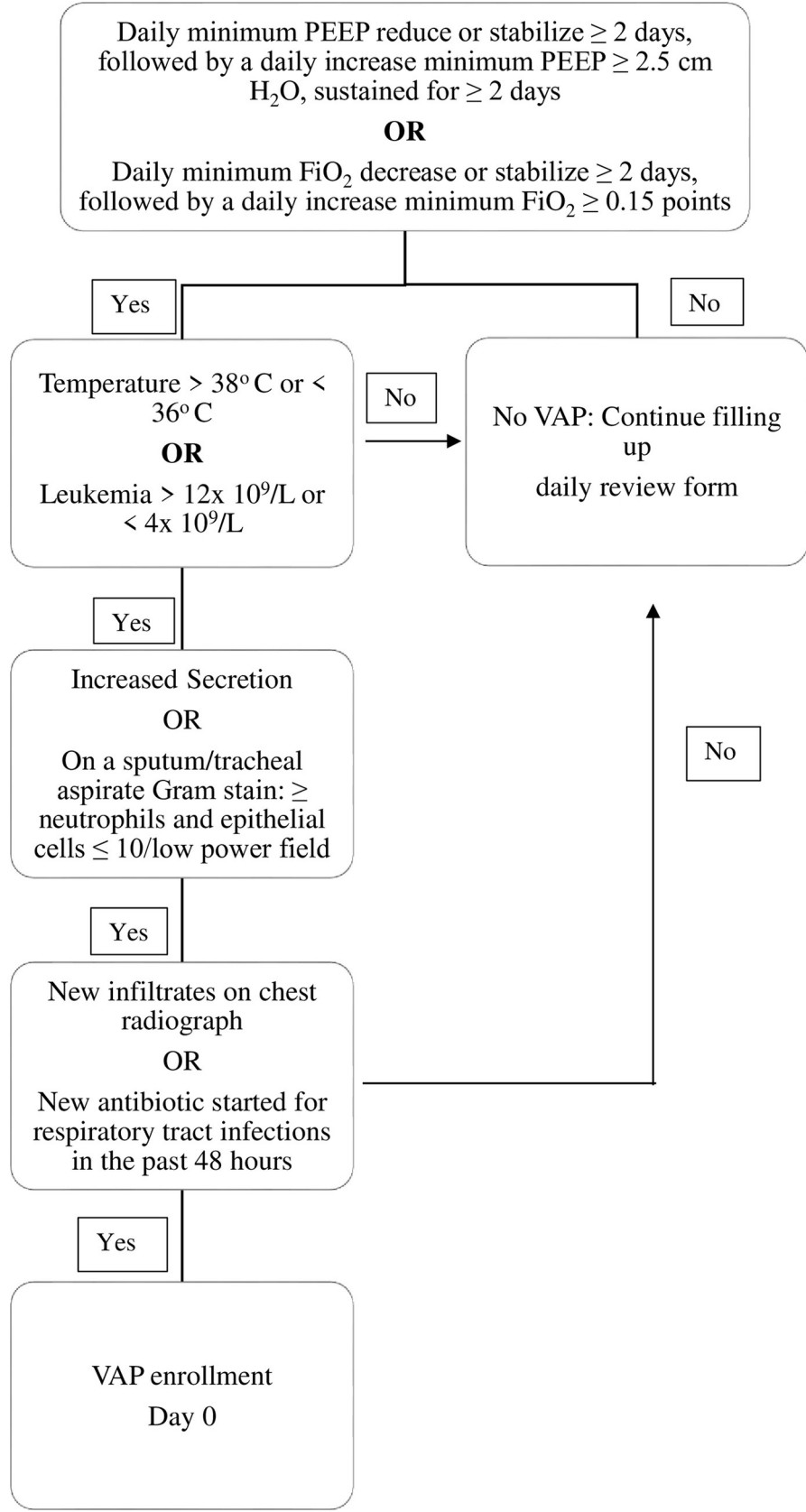

**Fig 1. Flow diagram for VAP definition.** Firstly, a deterioration in ventilation following a period of stability defined according to positive end expiratory pressure (PEEP): $\geq 2$ days of stable or decreasing daily minimum PEEP followed by a rise in daily minimum PEEP of $\geq 2.5$ cm $H_2O$, sustained $\geq 2$ calendar days; or $FiO_2$: $\geq 2$ days of stable or decreasing daily minimum fraction of inspired oxygen($FiO_2$) followed by a rise in daily minimum $FiO_2 \geq 0.15$ points, sustained $\geq 2$ calendar days. Secondly, systemic signs of fever >38˚C or <36˚C or white blood cell count >$12\times10^9$/L or <$4\times10^9$/L were required. Final criteria was an increased/new purulent tracheal aspirate (TA) samples/ tracheobronchial secretions or $\geq 25$ neutrophils per low power field (10 objective) on Gram stain of tracheal aspirate and either new and persistent infiltrates, consolidation, or cavitation as read by two study physicians on chest X-ray, or the decision to commence new antibiotic therapy.

using biochemical tests following standard operating protocol of Patan Hospital. In the interpretation of results, each colony corresponded to 20,000 CFU/ml, and it was considered to be TA positive when the count was $\geq 10^5$CFU/ml [26].

## Antimicrobial susceptibility testing

Antimicrobial susceptibility testing was performed using the Kirby Bauer disc diffusion method. The inhibitory zone sizes were interpreted according to the Clinical and Laboratory Standards Institute (CLSI) 2018 guidelines. Mueller–Hinton agar and antimicrobial discs were purchased from Mast Diagnostics, UK. *Escherichia coli* ATCC 25922 and *Staphylococcus aureus* ATCC 25923 were used as controls for these assays. The antimicrobials tested against *Acinetobacter* spp., *Pseudomonas* spp., and the Enterobacteriaceae were amikacin (30 mg), piperacillin/tazobactam (100/10 mg), imipenem (10 mg), ofloxacin (5 mg), and ceftriaxone (30 mg). An isolate was defined as MDR when it was non-susceptible to at least one agent in $\geq 3$ antimicrobial categories[CLSI guidelines (2018)] [27].Gram positive organisms were tested against co-trimoxazole (1.25/ 23.75 mg), penicillin (10 mg), gentamicin (10 mg), erythromycin (15mg) and oxacillin (1 mg).

## Statistical analysis

Data recorded onto a case record form were entered into a CliRes database system protecting participant information. Verification was done by double entry. Data analysis was performed in R Software (version 3.2). Categorical variables were expressed as count and percentage and analyzed using the Fisher's exact test. Continuous variables were expressed as median and interquartile range and analyzed using Kruskal-Wallis rank sum test and the pairwise Wilcoxon rank—sum test. Each variable with a p- value < 0.05 was considered a significant variable. VAP incidence was calculated as follows: (Number of cases with VAP/Total number of patients who received MVx100) = VAP rate per 100 patients. VAP incidence density was calculated as follows: (Number of cases with VAP/Number of ventilator days) x 1000 = VAP per 1,000 ventilator days. Flow diagram of the study enrollment procedure and categorization into five categories is shown in "Fig 2".

## Results

### Baseline characteristics

Four hundred and thirty-eight patients between April 2016 and March 2018were hospitalized in the ICU and enrolled in the study. The patients were between the ages of 18 and 95 years and 48.9% (214) were male and 51.1% (224) were female "Table 1".

The total numbers of participants in each of the five categories were I-P-V- = 111, I-P+V- = 122, I+P-V- = 12, I+P+V- = 152, and I+P+V+ = 41, who had median ages of 48, 62.5, 35.5, 57.5, and 59.5years, respectively. More than 50% of participants in all five categories had comorbidities at the time of admission in the ICU with >15% of the participants in each category

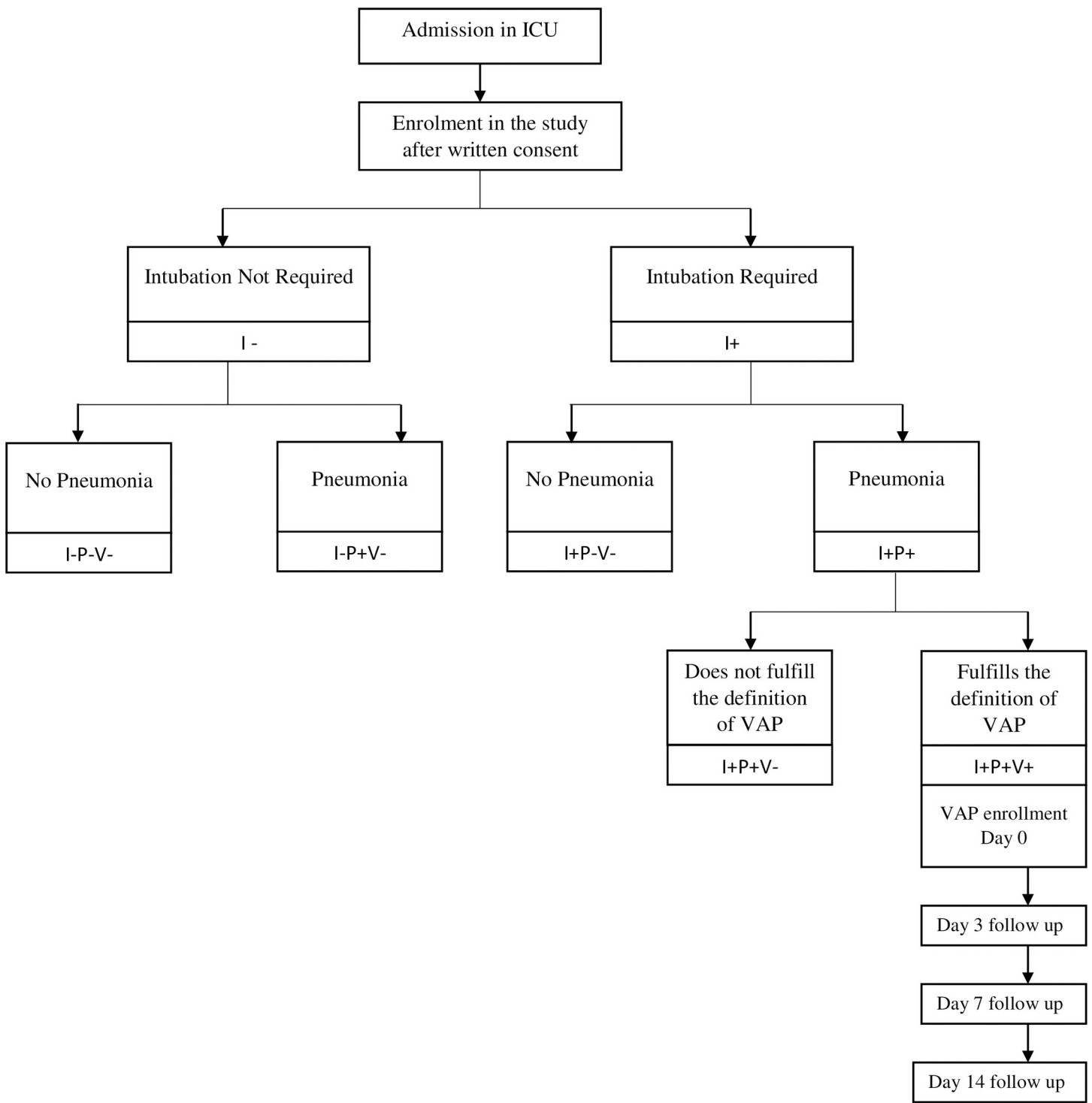

**Fig 2. Flow diagram of the study enrollment procedure and categorization into five categories.** Patients enrolled in the study were into five groups depending upon intubation, pneumonia development, and VAP development: These were: I-P-V- (not intubated, no pneumonia), I-P+V- (not intubated, but pneumonia developed), I +P-V- (intubated but no pneumonia), I+P+V-(intubated and pneumonia developed, but VAP not confirmed), and I+P+V+(VAP confirmed). The flow diagram of the study enrollment procedure and categorization into five categories is shown in "Fig 2".

**Table 1. Baseline characteristics of patients enrolled in the study.**

| Variables | I-P-V- (111) | I-P+V- (122) | I+P-V- (12) | I+P+V- (152) | I+P+V+ (41) |
|---|---|---|---|---|---|
| Age (years) | 48 (30–62) | 62.5 (48.5–72) | 35.5 (23.3–50.3) | 57.5 (35–72) | 59.5 (42.8–76.3) |
| Weight (Kilogram) | 59.5 (50–65) | 57 (49.5–64) | 60 (50.5–70.5) | 58 (50–65) | 58 (49–62) |
| M:F | 57:54 | 53:69 | 7:5 | 77:75 | 20:21 |
| SOFA | 3 (2–5) | 3 (2–5) | 6.5 (4–10) | 8 (5–11) | 10 (7–11) |
| APACHE | 10 (6–14) | 11 (7.3–14) | 12 (6.8–15.3) | 18 (13–24) | 17 (13–22) |
| Comorbidity | 60 (54.1%) | 90 (73.8%) | 7 (58.3%) | 91 (59.9%) | 27 (65.9%) |
| Hospital previous 90 days | 22 (19.8%) | 19 (15.6%) | 3 (25%) | 38 (25%) | 7 (17.1%) |
| Antibiotics in last 90 days | 11 (9.9%) | 28 (23%) | 3 (25%) | 54 (35.5%) | 4 (9.8%) |

Values given are median (IQR) or count (percent).

I-P-V- (Non intubated, no pneumonia), I-P+V- (non- intubated, pneumonia developed), I+P-V- (intubated, no pneumonia), I+P+V- (intubated, pneumonia but not VAP also called non–VAP), I+P+V+ (intubated, pneumonia and VAP confirmed).

having had a history of hospital admission in the past 90 days with antimicrobial use ranging from 9.8% (4/41 in the I+P+V+ group) to 35.5% (54/152 in the I+P+V- group). The median SOFA scores ranged from 3 to 10, with the highest scores being observed in the VAP group (median of 10, IQR 7–11). The median APACHE II score ranged from 10 to 18; the highest score was observed in the non-VAP group 18 (IQR 13–24).

## Association of different variables with intubation and pneumonia

In total, 29.7% (130/438) of patients had a diagnosis of pneumonia when admitted in the ICU and 46.8% (205/438) of patients required intubation. The most common requirement for intubation was failure to oxygenate (24.4%; 50/205), followed by failure to maintain or protect the airway (99%; 39/205). Pneumonia was common in both the intubated (94.14%; 193/205) and the non-intubated (52.36%; 122/233) patients. Pneumonia in intubated patients in the ICU was associated with longer days of stay (median of 10, IQR 5–15, $p < 0.001$) when compared to non-intubated patients with pneumonia (median of 4, IQR 3–6, $p < 0.001$). SOFA and APACHE II scores were also significantly higher among the patients that then on-intubated patients ($p < 0.05$; Kruscal-wallis rank sum test) "Table 2".

**Table 2. Kruskal-wallis rank sum test for the variables associated with intubation and pneumonia.**

| Variables | IntPosPneum | NonIntPosPneum | p—value |
|---|---|---|---|
| | I+P+V(V+and V-) | I-P+V- | |
| | (193) | (122) | |
| Days in Hospital | 11 (7–17) | 5 (3–7) | <0.001 |
| Days in ICU | 10 (5–15) | 4 (3–6) | <0.001 |
| Days in Intubation | 9 (5–15) | NA | NA |
| FiO2 | 50 (40–80) | 36 (29–41) | <0.001 |
| PaO2 | 79 (48.2–114) | 68.1 (53.7–88) | 0.056 |
| Temperature | 98 (97.2–99.2) | 98 (97.2–98.4) | 0.027 |
| SOFA | 8 (5–11) | 3 (2–5) | <0.001 |
| APACHE | 18 (13–23) | 11 (7.25–14) | <0.001 |
| Mortality | 86/193 (44.6%) | 13/122 (10.7%) | <0.001 |

Values given are median (IQR) or count (percent). Kruskal-Wallis rank sum test for median (IQR). Fisher's Exact Test for count (percent). I+P+ (intubated, pneumonia) and I-P+ (non- intubated, pneumonia developed).

**Table 3. Kruskal-Wallis rank sum test for the variables associated with different categories.**

|  | I-P-V- (n = 111) | I-P+V- (n = 122) | I+P-V- (n = 12) | I+P+V- (n = 152) | I+P+V+ (n = 41) | p—value |
|---|---|---|---|---|---|---|
| Days in Hospital | 4 (3–6) | 5 (3–7) | 5 (2.75–7) | 10 (7–14) | 18 (11–27) | <0.001 |
| Days in ICU | 3 (2–6) | 4 (3–6) | 4.5 (2.75–6) | 9 (5–12) | 16 (10–25) | <0.001 |
| Days in Intubation | NA | NA | 4 (2.75–7.25) | 7.5 (4–11) | 17 (11–27) | <0.001 |
| FiO2 | 29 (21–34) | 36 (29–41) | 40 (30–41) | 50 (40–70) | 60 (40–100) | <0.001 |
| PaO2 | 85 (65.8–111.4) | 68.1 (53.7–88) | 103 (64–150.5) | 82.5 (48.1–119) | 70.6 (52.3–102) | 0.041 |
| Temperature | 98 (97–98.7) | 98 (97.2–98.4) | 98.2 (97–99.2) | 98 (97.1–99.15) | 98.6 (98–99.2) | 0.085 |
| SOFA | 3 (2–5) | 3 (2–5) | 6.5 (4–10) | 8 (5–11) | 10 (7–11) | <0.001 |
| APACHE | 10 (6–14) | 11 (7.25–14) | 12 (6.75–15.25) | 18 (13–24) | 17 (13–22) | <0.001 |
| Mortality | 4/109(3.7%) | 13/122(10.7%) | 3/12 (25%) | 62/155 (40%) | 24/41 (58.5%) | <0.001 |

Values given are median (IQR) or count (percent). Kruskal-Wallis rank sum test for median (IQR).

Fisher's Exact Test for count (percent).

## Incidence density of VAP

Out of 205 patients requiring mechanical ventilation during their stay in ICU, 41 patients were diagnosed with VAP (20%), equating with a total incidence density of 16.45 cases per 1,000 ventilator days.

## Factors associated with VAP confirmed cases

We aimed to identify factors associated with VAP in the VAP confirmed group (I+P+V+) "Table 3". VAP was significantly associated with the duration of stay in the hospital (median 18 days, IQR 11–27, $p<0.001$), the duration of stay in ICU (median16 days, IQR 10–25, $p<0.001$), number of days of intubation (median 17 days, IQR 11–27, $p<0.001$), fraction of inspired oxygen(FiO2) (median 60, IQR 40–100, $p<0.001$), APACHE II score (median 17, IQR 13–22 $p<0.001$), SOFA score (median 10, IQR 7–11, $p<0.001$) and $PaO_2$ (median 70.6, IQR 52.3–102, $p<0.04$).

A pairwise Wilcoxon signed rank test was performed among the five defined categories to identify significance between the groups "Fig 3". The number of days of hospital stay, the number of days of ICU stay, APACHE II score, and SOFA score were all significantly higher in the I+P+V+ and I+P+V- groups than in the I-P-V- group ($p< 0.001$).

## Mortality

Mortality was significantly higher in patients with pneumonia requiring intubation than patients with pneumonia not requiring intubation (44.6% (86/193) vs. 10.7%, (13/122) $p<0.001$, Fisher's exact test) "Table 2". Between the groups, the highest mortality (58.5%; 24/41) was observed among VAP patients (I+P+V+) followed by non-VAP (I+P+V-) patients (40.8%,62/152); and lowest mortality (3.7%; 4/109) was observed among patients who neither required intubation nor had pneumonia during their stay in the ICU (I-P-V-)($p< 0.001$; Kruskal wallis test) "Table 3".

## Microbiology of VAP and non- VAP

A total of 110 samples from those with confirmed VAP(I+P+V+) and 130 samples from those without-VAP category (includes all categories except I+P+V+) were subjected to microbiological cultured. The majority of these samples were TA samples; 81/110 in VAP category and 63/130 in non-VAP category.

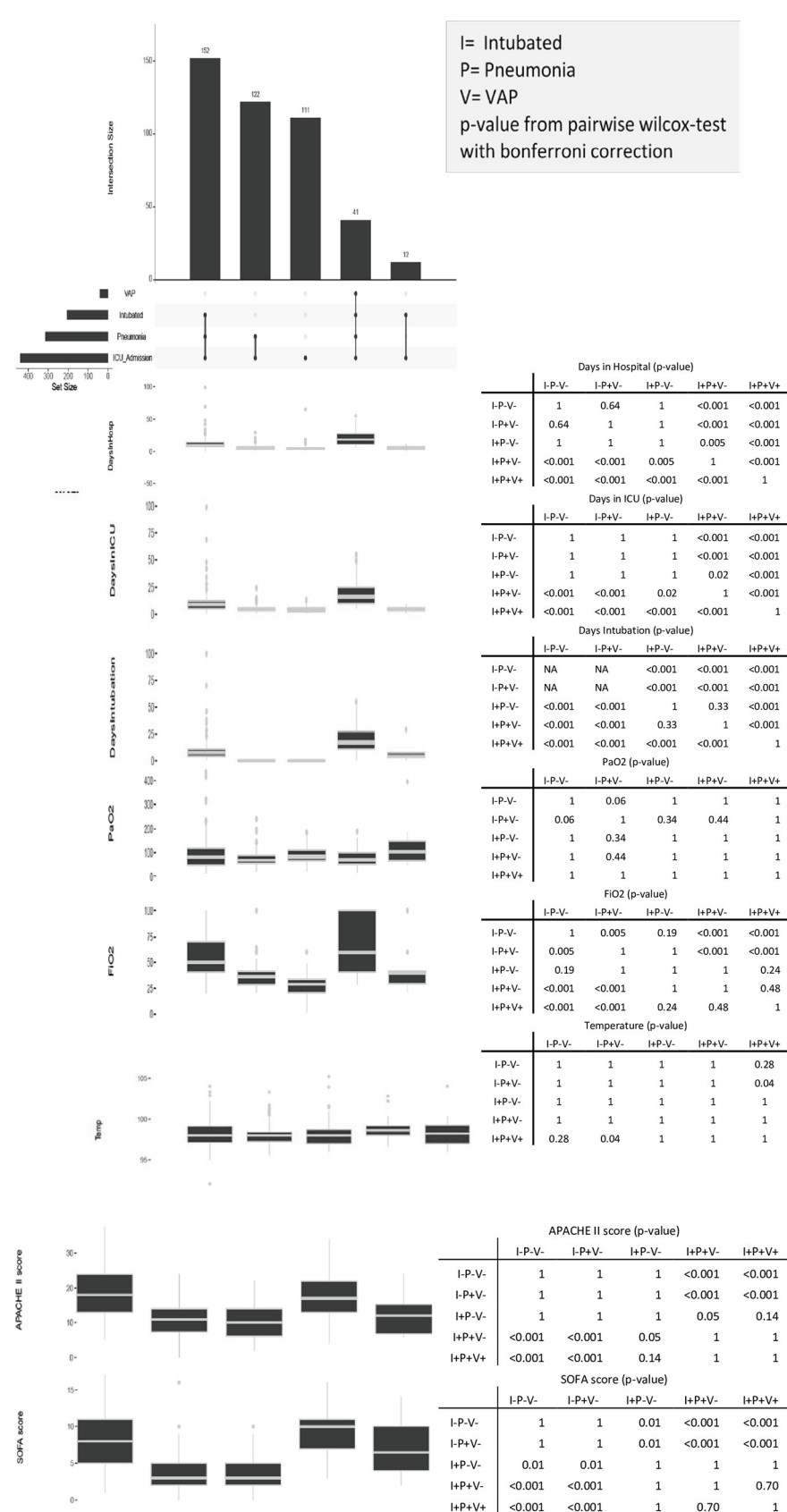

**Fig 3. Pairwise Wilcoxon signed rank test performed among the five defined categories to identify significance between the groups.** I = intubated, P = pneumonia, V = VAP. P-value from pairwise wilcox-test with bonferroni correction.

Gram negative bacteria were the dominant organisms from both VAP and non-VAP patients "Table 4", "Fig 4"). *Klebsiella* species was the most common bacteria associated with VAP (n = 36,32.7%) followed by *Acinetobacter* species (n = 35, 31.8%).*Acinetobacter* species was the predominant organism (n = 48, 36.9) isolated from those without-VAP, followed by *Klebsiella* species (n = 28, 21.5%).*Klebsiella* species was more likely to be isolated from VAP patients (OR1.76, 95%CI 0.96–3.3,p value 0.05).

## MDR in VAP and non-VAP categories

MDR was prevalent in all bacterial isolates from both VAP and non-VAP categories. The distribution of MDR isolates from various VAP and non-VAP samples are presented in "Fig 5". 90% (n = 99/110) of the isolates from various VAP samples and 81.5% (n = 106/130) of the non-VAP isolates were MDR. The data was suggestive of association of MDR with the VAP isolates but this was none significant (OR 2.03, 95%CI 0.90–4.85, p 0.07).

## Discussion

The data on HAP from prospective studies are scant notably from LMIC setting. Our study showed that HAP was common in our ICU setting regardless of intubation requirement [94.14% (193/205) among intubated and 52.36% (122/233) among non-intubated patients] indicating that these pneumonia cases may be a common HAI at Patan hospital. In addition to this, at least 16% of the patients from all the categories have had a visit to the hospital in the past 90 days and 50% of the patients had co-morbidities "Table 1" mainly chronic respiratory illness such as chronic obstructive pulmonary disorder. As a result, antibiotic usage was also common. Pneumonia as HAP was also common in a Malaysian study where,21% of HAP infections were pneumonia [6].

The complex interplay between the endotracheal tube, presence of risk factors, virulence of the invading bacteria and host immunity largely determine the development of VAP [28].The diagnosis of VAP is traditionally based on clinical symptoms and radiographic criteria that require further bacteriological confirmation. However, it has been demonstrated that these criteria are not sensitive or specific [8]. There is no gold standard for the diagnosis of VAP however, the qualitative method of culturing the tracheobronchial aspirate samples is said to be better at differentiating colonization and actual infection.

Despite recent advances in microbiological tools, the epidemiology and diagnostic criteria for VAP are still controversial, complicating the interpretation of treatment, prevention, and

**Table 4. Etiology of VAP and non-VAP specimens.**

| Isolates | VAP (n = 110) | Non-VAP (n = 130) | OR (95%CI) | p-value |
|---|---|---|---|---|
| *Acinetobacter* spp | 35 (31.8%) | 48 (36.9%) | 0.8 (0.45–1.4) | 0.42 |
| *Klebsiella* spp | 36 (32.7%) | 28 (21.5%) | **1.76 (0.96–3.3)** | **0.05** |
| *Pseudomonas* spp | 14 (12.7%) | 17 (13.1%) | 0.97 (0.42–2.21) | 1 |
| *E. coli* | 11 (10%) | 18 (13.8%) | 0.69 (0.28–1.64) | 0.43 |
| *Enterobacter* spp | 5 (4.5%) | 8 (6.2%) | 0.73 (0.18–2.61) | 0.78 |
| Coagulase negative *Staphylococcus* (CoNS) | 7 (6.4%) | 3 (2.3%) | 2.9 (0.6–17.6) | 0.19 |
| *S. aureus* | 0 (0%) | 5 (3.8%) | NA | NA |

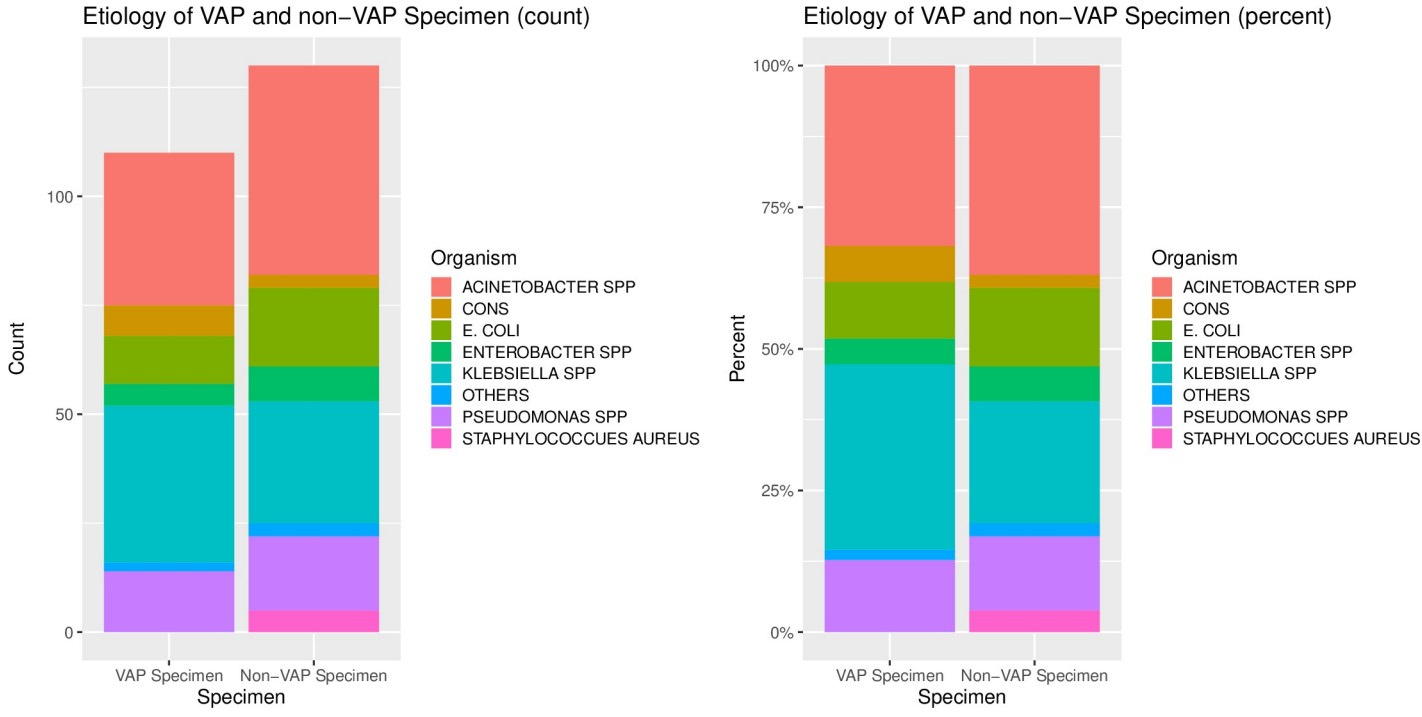

**Fig 4. Etiology of VAP and non-VAP specimen.**

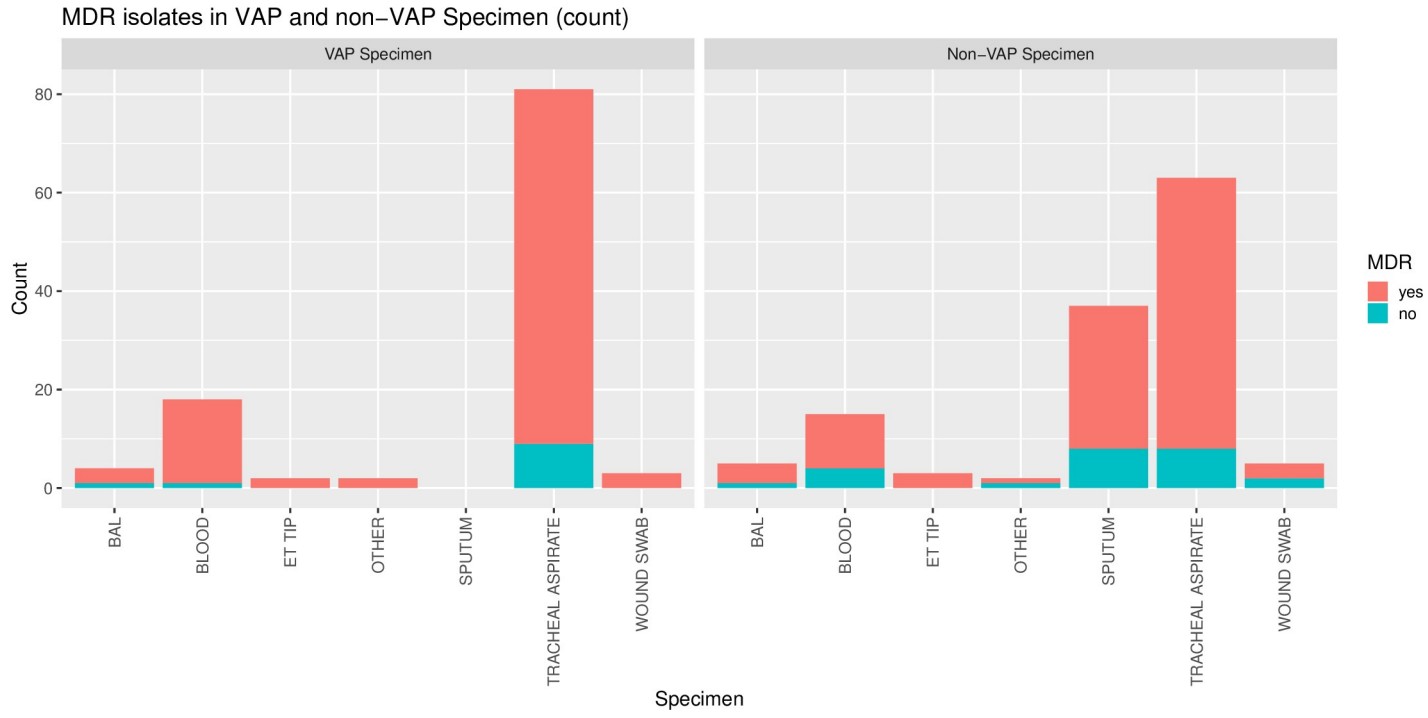

**Fig 5. MDR isolates in VAP and non-VAP specimen.**

outcomes studies [9]. Data on rates of VAP, the common associated pathogens, and their anti-microbial susceptibility profiles from Asia are limited [12]. Pooled incidence density of VAP was 18.5 per 1000 ventilator days in high in Asian LMIC countries [12]. This finding is similar to our study where we observed the incidence rate of VAP of 20% (n = 41/205) with a total of incidence density of 16.45 per 1000 ventilator days. However, some other Asian countries have reported a lower VAP incidence (9.9%) and VAP density (8.7/1000 ventilator days) [16]. Meta- analysis study from mainland China reported that the cumulative incidence of VAP was 23.8% [29]. In contrast, studies from India have reported the higher incidence density of VAP of 39.6% to 40.1% [30, 31].Such reported incidences vary widely from 5 to 40% depending on the setting and diagnostic criteria [9] indicating that the incidence rates vary not only between countries but also among different settings within a country. In high-income countries, a com-bination of surveillance, education, and tailored intervention and prevention bundles have led to a reduction in the incidence of VAP [32].

Gram negative bacteria were the dominant organisms from both VAP and non-VAP cate-gories. *Klebsiella* species was the most common bacteria associated with VAP followed by *Aci-netobacter* species Among non-VAP category, *Acinetobacter* species was the predominant organism followed by *Klebsiella* species (n = 28, 21.5%). *Klebsiella* species was more likely to be isolated from VAP category). We did not find major differences in the etiologic agents of VAP and non-VAP organism and their antimicrobial susceptibility profiles. However, there was a suggestive association of MDR with VAP isolates. Similar findings have been reported from other studies where the leading pathogens are *A. baumannii*, *P. aeruginosa* and *K. pneu-moniae* [33, 34]. In a large meta-analysis of 88 studies analyzing VAP in adults in Asia, it was revealed that *A. baumannii* was the most common organism in the LMIC group and the pro-portion due to this organism gradually reduced as income levels increased, and *S. aureus* and *P. aeruginosa* were the most common in the high income country group [12].Studies on VAP from other Asian countries also have reported *A. baumannii* to be the most common isolate [35].

One of the differences between our data and reports from Western countries was the pro-portion of gram- negative and gram-positive bacterial causes of VAP. We found a much lower proportion of gram-positive organisms as a causative agents of VAP and non VAP [35, 36].

ICUs often have the highest levels of infections due to antimicrobial resistant pathogens as a result of the environment that is under constant pressure with high antimicrobial usage due to the presence of severely ill patients. Etiologic agents of VAP are generally associated with pathogens with high levels of antimicrobial resistance, resulting in the need to treat with broad-spectrum antibiotics, which further drives antibiotic resistance [12].

Early onset VAP is usually attributed to antibiotic sensitive pathogens whereas late onset VAP is more likely caused by MDR bacteria and emerges after 4 days of intubation [37, 38]. However, this scenario seems to be different in the LMIC settings. This study revealed that MDR isolates were slightly higher in VAP than in non-VAP categories. Although significant association was not observed, there was an indication of association of VAP with MDR organ-isms. This further highlights the need to have infection control protocol guidelines in order to control such HAIs. s Guidelines for VAP prevention, including hand washing, elevation of the head of the bed, oral care with chlorhexidine, optimized endotracheal tube cuff pressure, respi-ratory circuit manipulation, and weaning protocols to early extubation were established in our hospital. These are cost effective control and preventive measures of VAP. Strict compliance, staff training, and regular monitoring of implementation of such guidelines will be effective in the prevention of VAP.

Mortality attributable to HAP is estimated between 5 and 13% [39].Even in HAP, generally considered to be less severe than VAP, serious complications occur in approximately 50% of

patients [40].Mortality was significantly higher in patients with pneumonia requiring intubation (44.6%, 86/193) than patients with pneumonia without intubation (10.7%, 13/122, p< 0.001). Highest mortality of 58.5% (24/41) was observed among VAP patients (I+P+V+) followed by non VAP (I+P+V-) patients (40.8%,62/152) and lowest mortality of 3.7% (4/109) was observed among patients who neither required intubation nor had pneumonia during their stay in the ICU (I-P-V-) category. The mortality in our ICU due to VAP was still lower than reported in other studies where it was as high as 68.4% [31].

Development of pneumonia increased the number of days of intubation. Intubated patients with pneumonia had to spend a median of 7.5(4–11) (I+P+V-) to 17(11–27) (I+P+V+) days being intubated as in comparison to those without pneumonia (I+P-V-) {4(2.75 to 7.25)}. In addition, VAP confirmed patients spent a median of16 days in ICU ranging from 10 to 27 days.24.6% (50 / 203) patients required intubation due to the reduction in exchange of oxygen (low PaO2), followed by cognitive impairment and airway obstruction (19.2%, 39/203).

## Conclusion

Pneumonia was one of the common infections in our ICU setting. Pneumonia developed among intubated patients in the ICU had longer days of stay in the ICU when compared to non-intubated patients with pneumonia. We found high VAP incidence in this study and highest mortality was observed among VAP patients followed by non VAP (I+P+V-) patients. MDR Gram negative bacteria were the dominant organisms from both VAP and non-VAP categories.

HAP including VAP remains to be the most prevalent HAIs at Patan hospital. One of the limitations of this study was that it was conducted at a single hospital. Surveillance studies on HAIs at various hospitals within the country are required in identifying the etiological agents. Antimicrobial susceptibility profiles of the etiological agents and outcomes of HAP and VAP would be beneficial for setting more appropriate guidelines for management of such diseases. In addition, countries like Nepal lack proper protocols of infection control and implementation for minimizing such infections in the hospital. Therefore, a suitable surveillance programs should be implemented, analyzing differences in VAP rates between different ICUs, and evaluating potential therapeutic approaches, and prevention strategies.

## Supporting information

**S1 Dataset.**
(XLSX)

## Acknowledgments

We wish to acknowledge all Patan hospital ICU staffs for facilitating data collection and laboratory staffs: Rajendra Shrestha, Bijaya Laxmi Karanjit, and Bidya Laxmi Shrestha for performing microbiological part of the study. The authors wish to thank all the individuals who consented to participate in this study.

## Author Contributions

**Conceptualization:** Sabina Dongol, Abhilasha Karkey.

**Data curation:** Nhukesh Maharjan, Sarita Pyatha, Rajkumar K. C.

**Formal analysis:** Nhukesh Maharjan.

**Funding acquisition:** Abhilasha Karkey.

**Investigation:** Sabina Dongol, Sarita Pyatha, Rajkumar K. C.

**Methodology:** Sabina Dongol, Sarita Pyatha, Abhilasha Karkey.

**Project administration:** Abhilasha Karkey.

**Supervision:** Sabina Dongol, Gyan Kayastha, Rajkumar K. C., Stephen Baker, Abhilasha Karkey.

**Writing – original draft:** Sabina Dongol, Nhukesh Maharjan.

**Writing – review & editing:** Sabina Dongol, Gyan Kayastha, Louise Thwaites, Buddha Basnyat, Stephen Baker, Abhilasha Karkey.

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
