## [Decision Letter · Decision Letter 0]

4 Aug 2021

PONE-D-21-22640

Epidemiology, etiology, and diagnosis of health care acquired pneumonia including ventilator-associated pneumonia in Nepal

PLOS ONE

Dear Dr. Karkey,

Thank you for submitting your manuscript to PLOS ONE. After careful consideration, we feel that it has merit but does not fully meet PLOS ONE’s publication criteria as it currently stands. Therefore, we invite you to submit a revised version of the manuscript that addresses the points raised during the review process.

Please, revise the manuscript paying special attention to the various suggestions provided by reviewer 2 and resubmit as early as your convenience

We look forward to receiving your revised manuscript.

Kind regards,

Monica Cartelle Gestal, PhD

Academic Editor

PLOS ONE

Journal Requirements:

2. In your ethics statement, please specify the date of your ethics approval.

Reviewers' comments:

Reviewer's Responses to Questions

**Comments to the Author**

1. Is the manuscript technically sound, and do the data support the conclusions?

Reviewer #1: Yes

Reviewer #2: Yes

2. Has the statistical analysis been performed appropriately and rigorously? 

Reviewer #1: Yes

Reviewer #2: Yes

3. Have the authors made all data underlying the findings in their manuscript fully available?

Reviewer #1: Yes

Reviewer #2: Yes

4. Is the manuscript presented in an intelligible fashion and written in standard English?

Reviewer #1: No

Reviewer #2: No

5. Review Comments to the Author

Reviewer #1: All abbreviations in the text must be spelled correctly (first in full).

English grammar needs to be revised.

CFU / ml must be spelled correctly in the text.

The sample was collected between 2016-2018, but the methods used CLSI 2016, this is a problem.

What is the meaning of CONS(coagulase-negative staphylococci)? Please refer to it in Table 5.

In the first paragraph of the discussion, it is better to first explain about health care acquired pneumonia.

Reviewer #2: The study is very interesting ; results and conclusion are very useful mainly in this part of world with low resources setting. Authors have clearly discussed pneumonia among patients in ICU; compared according to intervention done. Simliarly, VAP incidence, mortality, etiological agents , Antimicrobial susceptibility tests and other clinical profiles were the items thoroughly discussed in the study. The author also suggests different control measures and preventive strategies. Overall, the paper is good quality but some shortcomings need to be addressed

In abstract

Only add about the main aim of study not all and present in the results serially.

Introduction

The authors have put excellent effort to justify the rationale and background behind the problem.

Methods

The authors have done well in describing methodology under different headings. It would be great if you could further clarify inclusion and exclusion criteria.

Results

The authors have documented the results well mainly for association of intubation and pneumonia.

Discussion

It is good that you have tried to compare data of Nepal mainly VAP density with other Asian countries and LMICs. It would be nice to elaborate causes for contrasting results among these countries??

Why contrasting findings were seen among your study and other western countries study. What might be the likely cause for low numbers of gram positive organism?

As Nepal is poor in economy, please add some cost-effective control and preventive measures of HAP and VAP.

As the outcomes are relevant , possible description of risk factors of VAP in country like Nepal can also be included.

Are ventilator care bundles used in your study site?

Please add any strengths and limitations above conclusion in your study if any.

Minor issues

Spacing not done in some places like line no. 58

Referencing no. Should be after full-stop.

In table 1, write unit of weight , age

Page 266 to 230 n"=" is missing

And also Page 277 to 280 ; and other lines.

Brackets are only open or only closed some where.

Table 4 and 5 can be merged into a single table.

Correct typographical errors.

Thank you!

6. PLOS authors have the option to publish the peer review history of their article (what does this mean?). If published, this will include your full peer review and any attached files.

Reviewer #1: No

Reviewer #2: No

---

## [Author Response · Author response to Decision Letter 0]

4 Oct 2021

Answer: Correction made

2. In your ethics statement, please specify the date of your ethics approval.

Answer: Date added : NHRC approval date: 11March 2016 and OxTREC approval date was 19 Oct 2018. The study was started after obtaining the approval from Nepal health Research Council (NHRC).

Answer: Dataset is submitted as S-Dataset_xlsx. relevant URLs, DOIs, or accession numbers within your revised cover letter.

Reviewers' comments:

Reviewer's Responses to Questions

Comments to the Author

1. Is the manuscript technically sound, and do the data support the conclusions?

Reviewer #1: Yes

Reviewer #2: Yes

2. Has the statistical analysis been performed appropriately and rigorously?

Reviewer #1: Yes

Reviewer #2: Yes

3. Have the authors made all data underlying the findings in their manuscript fully available?

Reviewer #1: Yes

Reviewer #2: Yes

4. Is the manuscript presented in an intelligible fashion and written in standard English?

Reviewer #1: No

Reviewer #2: No

5. Review Comments to the Author

Reviewer #1: All abbreviations in the text must be spelled correctly (first in full).

English grammar needs to be revised.

CFU / ml must be spelled correctly in the text.

Answer: Corrected on the line 150 CFU/ml. The full form of CFU has been added to abbreviation list.

The sample was collected between 2016-2018, but the methods used CLSI 2016, this is a problem.

Answer: we corrected it to 2018. We checked the interpretation of the antibiotics that we tested against 2016 and 2018 CLSI guidelines and confirm that both have the same interpretation criteria for the antibiotics that we tested in this study.

What is the meaning of CONS(coagulase-negative staphylococci)? Please refer to it in Table 5.

Answer: CONS has been corrected as CoNS. The full form is coagulase negative Staphyloccus. Full form is added in the table 5.

In the first paragraph of the discussion, it is better to first explain about health care acquired pneumonia.

Answer: The data on HAP from prospective studies are scant notably from LMIC setting. This is added in the discussion

Reviewer #2: The study is very interesting ; results and conclusion are very useful mainly in this part of world with low resources setting. Authors have clearly discussed pneumonia among patients in ICU; compared according to intervention done. Simliarly, VAP incidence, mortality, etiological agents , Antimicrobial susceptibility tests and other clinical profiles were the items thoroughly discussed in the study. The author also suggests different control measures and preventive strategies. Overall, the paper is good quality but some shortcomings need to be addressed

In abstract

Only add about the main aim of study not all and present in the results serially.

Answer: We included only the main aim of the study in the abstract. This was followed by the summary of the statistical analysis.

Introduction

The authors have put excellent effort to justify the rationale and background behind the problem.

Methods

The authors have done well in describing methodology under different headings. It would be great if you could further clarify inclusion and exclusion criteria.

Answer: There were no exclusion criteria per se. One line was added in the study structure section: Patients who denied consent and patients under the age of 18 years of age were not included in the study.

Results

The authors have documented the results well mainly for association of intubation and pneumonia.

Discussion

It is good that you have tried to compare data of Nepal mainly VAP density with other Asian countries and LMICs. It would be nice to elaborate causes for contrasting results among these countries??

Answer: In high-income countries, a combination of surveillance, education, and tailored intervention and prevention bundles have led to a reduction in the incidence of VAP

Why contrasting findings were seen among your study and other western countries study. What might be the likely cause for low numbers of gram positive organism?

Answer: We are unable to provide explanation based on the findings from the study. There could be multidimensional factors ranging from patient care management protocols, antibiotic stewardship program, VAP bundle implementation, and continuous training and surveillance system.

As Nepal is poor in economy, please add some cost-effective control and preventive measures of HAP and VAP.

Answer: Guidelines for VAP prevention, including hand washing, elevation of the head of the bed, oral care with chlorhexidine, optimized endotracheal tube cuff pressure, respiratory circuit manipulation, and weaning protocols to early extubation were established in our hospital. These are cost effective control and preventive measures of VAP. Strict compliance, staff training, and regular monitoring of implementation of such guidelines will be effective in the prevention of VAP. This is added in the discussion

As the outcomes are relevant , possible description of risk factors of VAP in country like Nepal can also be included.

Answer: We have analysed data by grouping them into categories of intubation required and non intubated group and further divided into five categories into VAP development. We measured various variables and their association to each group.

Are ventilator care bundles used in your study site?

Answer: Guidelines for VAP prevention, including hand washing, elevation of the head of the bed, oral care with chlorhexidine, optimized endotracheal tube cuff pressure, respiratory circuit manipulation, and weaning protocols to early extubation were established in our hospital. These are cost effective control and preventive measures of VAP. Strict compliance, staff training, and regular monitoring of implementation of such guidelines will be effective in the prevention of VAP.

Please add any strengths and limitations above conclusion in your study if any.

Answer: These lines in the conclusion section explains the strength of this study.

Pneumonia was one of the common infections in our ICU setting. Pneumonia developed among intubated patients in the ICU had longer days of stay in the ICU when compared to non-intubated patients with pneumonia. We found high VAP incidence in this study and highest mortality was observed among VAP patients followed by non VAP (I+P+V-) patients. MDR Gram negative bacteria were the dominant organisms from both VAP and non-VAP categories. 

One of the limitations of this study was that it was conducted at a single hospital. Surveillance studies on HAIs at various hospitals within the country are required in identifying the etiological agents. Antimicrobial susceptibility profiles of the etiological agents and outcomes of HAP and VAP would be beneficial for setting more appropriate guidelines for management of such diseases.

Minor issues

Spacing not done in some places like line no. 58

Answer: Corrected

Referencing no. Should be after full-stop.

Answer: we referred to the Plos one formatting style.

In table 1, write unit of weight , age

Answer: Corrections were made. Added kilogram as a unit of weight and Years along with age in table 1.

Page 266 to 230 n"=" is missing

And also Page 277 to 280 ; and other lines.

Answer: Corrected throughout the manuscript.

Brackets are only open or only closed some where.

Answer: Checked throughout the manuscript and corrected

Table 4 and 5 can be merged into a single table.

Answer: We have removed table 4 as this information is depicted in Fig 5. We are keeping Table 5 as it is (without merging it to table 4).

Correct typographical errors.

Answer: Done

Thank you!

6. PLOS authors have the option to publish the peer review history of their article (what does this mean?). If published, this will include your full peer review and any attached files.

Do you want your identity to be public for this peer review? For information about this choice, including consent withdrawal, please see our Privacy Policy.

Reviewer #1: No

Reviewer #2: No

---

## [Decision Letter · Decision Letter 1]

25 Oct 2021

Epidemiology, etiology, and diagnosis of health care acquired pneumonia including ventilator-associated pneumonia in Nepal

PONE-D-21-22640R1

Dear Dr. Karkey,

We’re pleased to inform you that your manuscript has been judged scientifically suitable for publication and will be formally accepted for publication once it meets all outstanding technical requirements.

Kind regards,

Monica Cartelle Gestal, PhD

Academic Editor

PLOS ONE

Additional Editor Comments (optional):

Reviewers' comments:

Reviewer's Responses to Questions

**Comments to the Author**

1. If the authors have adequately addressed your comments raised in a previous round of review and you feel that this manuscript is now acceptable for publication, you may indicate that here to bypass the “Comments to the Author” section, enter your conflict of interest statement in the “Confidential to Editor” section, and submit your "Accept" recommendation.

Reviewer #2: All comments have been addressed

2. Is the manuscript technically sound, and do the data support the conclusions?

Reviewer #2: Yes

3. Has the statistical analysis been performed appropriately and rigorously? 

Reviewer #2: Yes

4. Have the authors made all data underlying the findings in their manuscript fully available?

Reviewer #2: Yes

5. Is the manuscript presented in an intelligible fashion and written in standard English?

Reviewer #2: Yes

6. Review Comments to the Author

Reviewer #2: The authors have address all the queries. Congratulations! on publication og article. This article address major issue in developing countries like Nepal.

7. PLOS authors have the option to publish the peer review history of their article (what does this mean?). If published, this will include your full peer review and any attached files.

Reviewer #2: **Yes: **Sanjeev Kharel

---

## [Editor Report · Acceptance letter]

8 Nov 2021

PONE-D-21-22640R1 

Epidemiology, etiology, and diagnosis of health care acquired pneumonia including ventilator-associated pneumonia in Nepal 

Dear Dr. Karkey:

I'm pleased to inform you that your manuscript has been deemed suitable for publication in PLOS ONE. Congratulations! Your manuscript is now with our production department. 

Kind regards, 

on behalf of

Dr. Monica Cartelle Gestal 

Academic Editor

PLOS ONE